# Nurse Reports of Stressful Situations during the COVID-19 Pandemic: Qualitative Analysis of Survey Responses

**DOI:** 10.3390/ijerph17218126

**Published:** 2020-11-03

**Authors:** Judith E. Arnetz, Courtney M. Goetz, Bengt B. Arnetz, Eamonn Arble

**Affiliations:** 1Department of Family Medicine, College of Human Medicine, Michigan State University, Grand Rapids, MI 49503, USA; goetzcou@msu.edu (C.M.G.); arnetzbe@msu.edu (B.B.A.); 2Department of Psychology, Eastern Michigan University, Ypsilanti, MI 48197, USA; earble2@emich.edu

**Keywords:** nursing, occupational health, COVID-19, work stress, content analysis, qualitative

## Abstract

The coronavirus disease (COVID-19) pandemic has exposed nurses to conditions that threaten their health, well-being, and ability to work. It is therefore critical to study nurses’ experiences and well-being during the current crisis in order to identify risk groups for ill health and potential sources of organizational intervention. The aim of this study was to explore perceptions of the most salient sources of stress in the early stages of the coronavirus pandemic in a sample of U.S. nurses. A cross-sectional online survey was conducted among a sample of 695 U.S. nurses in May 2020. Content analysis was conducted on nurses’ responses (n = 455) to an open-ended question on the most stressful situations they had experienced during the pandemic. Six distinct themes emerged from the analysis: exposure/infection-self; illness/death-others; workplace; personal protective equipment/supplies; unknowns; opinions/politics. Two sub-themes concerned restrictions associated with the pandemic and feelings of inadequacy/helplessness regarding patients and their treatment. More than half of all comments concerned stress related to problems in workplace response to the pandemic. Healthcare institutions should provide opportunities for nurses to discuss the stress they are experiencing, support one another, and make suggestions for workplace adaptations during this pandemic.

## 1. Introduction

The novel coronavirus disease (COVID-19) pandemic has presented an unprecedented challenge to healthcare systems across the globe [1]. The rapid spread of the disease in late 2019 and early 2020 caught many healthcare systems off guard and scrambling to provide intensive care unit beds, ventilators, and personal protective equipment (PPE) for both healthcare workers and patients. With the pandemic, nurses have confronted a perfect storm of conditions that threaten their health, well-being, and ability to perform their jobs [2]. Media reports from many of the world’s COVID hotspots, including Italy [3,4] and the United States [5,6], document extreme exhaustion, physical discomfort from long working hours with face masks and other PPE, fear of contagion, and emotional distress in nurses. This combination of physical and emotional strain on an already stressed nursing workforce [2] has become a hallmark of the COVID-19 pandemic [7,8]. It is therefore critical to study nurses’ experiences and well-being during and in the aftermath of the current crisis in order to identify risk groups for ill health and potential sources of organizational intervention.

Nurses’ reactions to the stress of the current pandemic must be viewed from an occupational health and safety perspective. Stress and burnout were recognized internationally as work hazards for nurses [9] before the pandemic. Although research suggests that both occupational and personality factors play a role in burnout [10,11], in 2019 the World Health Organization declared burnout an occupational phenomenon—rather than a medical condition [12]. Characterized by feelings of exhaustion, disengagement from one’s job, and a sense of diminished professional fulfilment, burnout is considered the result of chronic work stress that the individual is not able to manage [12]. The onset of the COVID-19 pandemic has increased work stress among an already strained nursing corps, putting their mental health and well-being at risk [13,14,15]. Recent research from China [16,17,18] and Italy [19], two nations that experienced the early phase of the pandemic, found that nurses directly involved in the care of COVID-19 patients were at increased risk for mental health problems compared to other healthcare professionals. These early papers concerning the pandemic’s impact on healthcare workers pinpointed frontline exposure to COVID patients as a main risk factor but identified few other variables explaining the reported symptoms of depression, anxiety, insomnia [17,19], psychological distress [17], and post-traumatic stress [19] among nurses. The Italian study of 1379 healthcare workers, of which 472 of were nurses, did find that having a colleague who was hospitalized or quarantined was associated with worse mental health, but those results were not examined by professional group [19].

During the period 20 March–10 April 2020, the American Nurses Association (ANA) conducted a nationwide survey of nurses’ concerns and experiences during the early phase of the pandemic in the United States. A total of 32,000 nurses responded to the survey that asked about their professional needs and concerns related to the COVID outbreak. Nearly three quarters (74%) reported that their primary concern was lack of adequate personal protective equipment (PPE) and more than two thirds (64%) were concerned for the safety of their family and friends. More than 85% expressed fear of going to their workplace [20]. Although large in scale, the survey focused on perceived concerns and needs regarding education and staffing and did not pose questions about physical and mental health or stressful situations. Two recent qualitative studies explored the experiences of nurses’ [21] and both nurses and physicians [22] in the early stages of the pandemic in China. Sun et al. [21] focused on 20 nurses’ psychological responses, reporting that initial negative emotions evolved into a mixture of positive and negative emotions. That study identified fatigue, fear, and anxiety related to the unknown as sources of negative emotions but did not identify other specific causes of stress. Q. Liu and colleagues [22] interviewed nine nurses and four physicians from a province in China about their experiences caring for COVID-19 patients. That study identified three main themes: a feeling of duty/responsibility, work-related challenges, and resilience amid the challenges. However, both these studies are limited by small sample sizes from a single hospital [21] and province [22], thus limiting the generalizability of the studies’ findings.

While research on the effects of the pandemic on nurses’ health and well-being is still sparse, several recent editorials highlight a number of stressful factors that could potentially contribute to mental health problems [2,23,24,25]. These include fear of infection for oneself and one’s loved ones [2,23], the high rates of disease transmission and fatality [2], but also fear of the unknown regarding this disease [23]. Levels of work-related stress and the burden of extremely long working hours are recurrent topics [2,24] that are also reflected in research from China [21,22,26]. Nurses are also experiencing higher than normal patient-nurse ratios and many have been deployed to work outside of their specialty disciplines [24], factors that add to the stress of contagion, shortage of appropriate PPE, and fear of the unknown [2]. Several of the editorials note that nurses are experiencing levels of patient deaths that are unprecedented, even within a profession where encountering death and dying is expected [23,24]. One editorial described this as a virtual “tsunami of death” [23] (p. 2042) that combined with work exhaustion can lead to strong feelings of professional failure even among experienced nurses [23]. Several media reports corroborate these editorials, including the additional strain of repeatedly informing patients’ family members unable to be with the patient about the death of their loved one [6,8].

A recent survey examined nurses’ perceptions of working during the early stages of the pandemic in the U.S. and found that more than 50% of respondents experienced symptoms of depression and anxiety and close to one-third had symptoms of post-traumatic stress disorder. Lack of adequate PPE was a significant risk factor for all three mental health outcomes [13]. While important, it is unlikely that lack of adequate PPE is the only factor, and that other experiences are likely impacting nurses’ mental health symptoms. A closer analysis of responses in that survey to an open-ended question on stressful situations would help to shed light on circumstances and conditions influencing nurses’ reactions to working during the pandemic. Qualitative research from previous pandemics emphasized the importance of studying the experiences of frontline nurses to inform effective workplace and national responses during future healthcare crises [27,28]. Better understanding of current conditions could potentially support organizational and workplace efforts to mitigate the stress and ill-health that nurses are experiencing during, and in the wake of, the pandemic.

## 2. Materials and Methods

### 2.1. Aim

The aim of this study was to explore perceptions of the most salient sources of stress in the early stages of the coronavirus pandemic in a sample of U.S. nurses.

### 2.2. Design

A cross-sectional survey study of nurses in Michigan was conducted in May 2020. The 85-item questionnaire was developed by the research team for the purpose of the study. It included measures regarding demographic and work-related factors, as well as COVID-19 experiences concerning patient contact, emergency preparedness, personal protective equipment, fear, and mental health and well-being [13]. In addition to the 84 forced-choice items, the final question of the survey was open-ended and asked nurses to describe “the most stressful situations you have dealt with during the COVID-19 pandemic”. In this study, qualitative content analysis was used to examine nurses’ responses to this question.

### 2.3. Participants

Participants were recruited from the Michigan chapter of the American Nurses Association (ANA), the Michigan Organization of Nurse Leaders (MONL), and the Coalition of Michigan Organizations of Nursing (COMON). All members of the organizations, along with their colleagues, were eligible to participate (approximately 18,300 nurses). During the survey period, 695 responses were collected. Of these, 455 nurses also responded to the open-ended question (65.47%) and it is these nurses’ responses that are examined in the current study. The majority of this sample was female (n = 429, 94.3%). Approximately 18.1% (n = 82) of respondents were younger than 35 years of age, 21.8% (n = 99) were 35–44, 24.8% (n = 113) were 45–54, 26.6% (n = 121) were 55–64, and 8.7% (n = 38) were 65 or older. Just over 12% (n = 57) had been employed as a nurse for less than five years, 18.5% (n = 84) for five to ten years, and 69.0% (n = 314) for ten years or more. The majority of respondents were registered nurses (RNs) (86.6%, n = 394), followed by advanced practice registered nurses (APRNs) (12.7%, n = 58). Most worked in an inpatient/hospital setting (52.7%, n = 240) or an outpatient setting (22.9%, n = 104). Approximately 18% (n = 80) of respondents reported never working with COVID-positive patients, while the remaining 82.4% (n = 375) had been in contact with COVID-positive patients at least once. 22.0% (n = 100) of participants reported being in contact with COVID-positive patients very often/daily. Nurses who responded to the survey’s open question did not differ significantly from non-respondents with regard to age, gender, managerial position, years of work experience, or current stress level (*p* > 0.05 for all variables). However, a significantly larger proportion of open question non-respondents worked in inpatient care (72.9%) compared to question respondents (*p* = 0.000).

### 2.4. Data Collection

ANA Michigan distributed surveys directly to nurse members. COMON and MONL used a snowball recruitment technique, asking their members to distribute the survey within their respective organizations. Each of the organizations sent a link to the online survey via email.

### 2.5. Ethical Considerations

Each nurse who agreed to participate completed a consent statement in Qualtrics before continuing to the survey questions. The survey was confidential and anonymous, and the participants could terminate their participation at any time. The study was determined exempt by the Institutional Review Board at Michigan State University (Study 00004459).

### 2.6. Data Analysis

The consolidated criteria for reporting qualitative research, COREQ [29], were followed in the planning and execution of this study to ensure methodological integrity. Qualitative content analysis of the open-ended responses was conducted using a data-driven inductive approach to code content into themes [30]. Thematic analysis provides a framework for structuring qualitative data by establishing a coding system in which codes are grouped into recurrent themes that are relevant to the research question [31]. Structuring the data in this way helps to create meaning out of complex raw data [32].

### 2.7. Theme Development

In order to identify themes in the nurses’ reports of stressful experiences, a researcher with training in qualitative analysis (CG) thoroughly examined the responses and noted recurring issues. This researcher then reexamined the responses with these preliminary themes in mind, and either eliminated or verified and defined them. These preliminary themes and their definitions were sent to a second researcher (EA), who then examined the responses and noted where themes needed further clarification and suggested themes to be added or removed. The two researchers discussed and refined the themes until agreement was reached. The final definitions were then recorded in a coding form.

### 2.8. Coding

Following the theme development, the two researchers (CG and EA) developed coding instructions. It was decided that coders could code responses for more than one theme, and could also leave responses blank if necessary. Secondhand information (i.e., “I heard ____,” “My friend said ____”) could be coded, and coders were instructed to use their best judgment as to whether a response fit into a theme, as the definitions were non-exhaustive.

At this point, the researchers independently coded the first 25 responses to test interrater reliability (IRR) and establish agreement using these theme definitions. Agreement was 94% and the theme definitions were therefore assumed to be reliable. The researchers then tested IRR for the first 100 and 200 responses, at which point the percent agreement stayed constant. The coders then completed coding for the entire sample, and the final percent agreement was 95.21%. A third researcher (JA), who was not involved in the development of the themes, examined the coding and made a final coding decision on any disagreements. This resulted in the identification of two subthemes, which were defined and coded by JA and CG.

### 2.9. Validity and Reliability/Rigour

Qualitative rigor was fulfilled using Lincoln and Guba’s criteria (creditability, transferability, dependability, and confirmability) as a guide [33]. Credibility, which is similar to internal validity in quantitative studies, was achieved by comprehensiveness in data collection and analysis. All three coders became highly familiar with the data by reading through the responses multiple times in order to achieve accurate coding. Transferability, analogous to external validity, was assured by using direct quotes to illustrate the results. Dependability (similar to reliability) was achieved by using one coder who was not involved in the development of the themes. Confirmability was achieved through analyst triangulation involving three researchers. All coders analyzed the verbatim responses, then validated findings amongst themselves. Source triangulation was also used, as responses were collected from nurses in a variety of settings.

## 3. Results

The analysis revealed six major themes: exposure/infection, illness/death, workplace, PPE/supplies, unknowns, and opinions/politics (Table 1).

### 3.1. Exposure/Infection

This theme comprised the fear of the self being exposed to COVID-19 and becoming ill, as well as actually being exposed and becoming ill. In addition, this theme could encapsulate one’s fear of passing the virus onto others, such as loved ones and patients. Examples of nurse responses include:


*“[T]he most stressful part about this is being in a clinic that is still open that isn’t necessarily aware of patients covid status. [W]e are coming in contact every day with our patients who may have it and we may not even know it. [T]hen we are going home to our families and potentially spreading it. I know that if I was the cause of a loved one of mine getting it I would never forgive myself for that…”*



*“I brought COVID 19 home (I had the virus) and infected my husband. He ended up with a bilateral pneumonia hospitalized. It is very stressful to think I could infect someone else as well”.*



*“Fear of catching COVID-19 and transmitting to aging parents, fear of death from increased risk of exposure at work…”*


#### Subtheme: Restrictions

One subtheme was identified within the main theme of exposure/infection. This subtheme encompassed the restrictions associated with the pandemic, such as social distancing and the closing of businesses. An example of a response coded into this subtheme follows:


*“I rely on my family and friends as a support system and not being able to visit with them in person really ways [sic] on me and affects me as a whole person”.*


### 3.2. Illness/Death

In contrast to the exposure/infection theme, this theme represents dealing with infection, illness, and death of others, usually patients, coworkers, or loved ones. This includes witnessing the rapid deterioration of patients, witnessing patient death, and caring for ventilated or extremely ill patients. This also includes stress related to keeping ill patients isolated from family members and loved ones. In addition, this theme includes concern for others becoming ill, such as coworkers at a different hospital, family members, and friends. Nurse responses include:


*“Watching families being separated from their loved ones, especially in end of life situations. Seeing the fear in the eyes of Covid patients that cannot breathe and are begging me not to let them die”.*



*“The most upsetting thing is to meet patients when they come into the hospital walking and talking and to get to know them personally. Then to be the same nurse just a short time later to take care of the same patient after their passing, by putting them in a body bag and wheeling them to the morgue”.*


#### Sub-Theme: Inadequacy

Coders identified one subtheme within the illness/death theme. This sub-theme captures feelings of inadequacy and helplessness during the pandemic, especially in relation to their patients’ condition and treatment. The following is an example of a response coded into the inadequacy subtheme:


*“Watching patient’s [sic] suffocate while intubated and having nothing else that I can do for them. We have exhausted all efforts and there is literally nothing left to do. It’s emotionally and mentally taxing as a health care professional to feel as though there is nothing more I can do to help the patient”.*


### 3.3. Workplace

This theme constitutes work-related problems. Examples include relationships with coworkers, perceived workplace administrative failings, and failure to provide supplies and training. This also includes being assigned a high load of extremely ill patients or being assigned to new departments without training. Examples of nurse responses coded into this theme include:


*“I have significant fear and lack of trust in workplace protections. I no longer trust my employer has my safety as a priority. They have twisted the truth and flat out lied to nurses. They failed, and continue to fail to provide adequate PPE and adequate training...”*



*“Poor support from leadership during the Pandemic. Poor direction and poor organization from leadership. The redeployment experience was the key factor in my poor mental/physical/emotional health status. Poor support for redeployment experience”.*



*“…[A]dmin keeps telling us how bad we are failing, but they have ‘Healthcare Heroes’ signs on the front law[n]”.*


### 3.4. PPE/Supplies

This theme encapsulates stressors related to PPE/supplies. This includes not having enough PPE, cleaning supplies, ventilators, and testing supplies. Other examples include having to re-wear PPE, unclear PPE guidelines, and physical discomfort related to wearing PPE. The following are examples of nurse responses related to PPE/Supplies:


*“Being denied PPE. I had to break a strap on a mask to receive a new one (that I had worn for 5 12 h shifts and sneezed in it) and my place of work punished me for it, calling me belligerent and damaging property”.*



*“Our floor is only provided an N95 if we have a droplet patient. Since the Covid floor has opened, we no longer see these droplet patients unless this error is occurring. I have the same N95 since March. We also are only allowed one surgical mask for one week. We still are rationing PPE…”*



*“The worst part of this is having appropriate PPE at home and not being able to bring it to work and use it. It feels like I am leaving my oxygen at home or something vital to life just because the hospital doesn’t want to look bad”.*


### 3.5. Unknowns

This theme encapsulates dealing with the constant unknowns during the epidemic. This includes changing understanding of COVID symptoms and PPE necessities. This also includes concerns about a potential surge, not knowing when the pandemic will end, as well as social isolation, childcare, and family-related issues. Finally, this theme can encompass concerns related to working conditions, job security and financial/economic security, including furloughs/layoffs. Examples of nurse responses include:


*“Right now dealing with husband being put on indefinite unpaid furlough and fear he will not be called back to work due to his age”.*



*“Recurring nightmares, not having answers for people, worrying about parents, students, spouse. Worried about being furloughed, money, job security everything!!!”*


### 3.6. Opinions/Politics

This theme encapsulates family/community opinions related to COVID-19, as well as the politicization of the pandemic. This includes dealing with perceived failings of state/federal administration, seeing false information spread, and dealing with protestors/people who believe COVID-19 is a hoax. This theme also includes negative judgment and fear displayed toward nurses related to COVID-19. Examples of responses coded into this theme include:


*“The most stressful thing is the angry and hurtful things people say who don’t believe Covid-19 is a thing. That is the thing that bothers me the most. Also, the lack of compassion and empathy people have for one another. It’s exhausting trying to defend the seriousness of this virus. I cannot go on social media without crying due to all the ignorance and hatred”.*



*“...Most recently, my stress/anger has been related to the politics surrounding COVID-19. To me, stay at home orders and closed businesses is not political-this is a global crisis due to a deadly and virulent virus, but because it has become so politically charged, it makes it hard to voice opinions and concerns related to government orders/decisions, even at work. …”*


## 4. Discussion

The aim of this study was to explore perceptions of the most salient sources of stress in a sample of U.S. nurses during the early stages of the coronavirus pandemic. More than two-thirds of questionnaire respondents (n = 455, 65.47%) responded to the single open-ended survey item. This is unusually high in studies of this nature, as non-response on open-ended questions is a common and recognized disadvantage in questionnaires [34]. This speaks to the perceived relevance of the question of pandemic-related stress to the nurses in our sample. Many of them also wrote lengthy responses that covered several of the identified themes.

All six themes identified in our analysis have been mentioned to some degree in media reports (e.g., [5,6]) and opinion pieces in nursing journals (e.g., [23]) and were in many ways not surprising. Fear of becoming infected by a potentially lethal virus is an understandable and expected source of stress. Considering the highly contagious nature of the severe acute respiratory syndrome (SARS) coronavirus-2 causing COVID-19, nurses with frequent contact with infected or potentially-infected patients are understandably anxious about contracting the illness themselves and passing it on to others [2,13,35]. It is interesting to note that subthemes were identified for both the exposure/infection to self theme and the illness/death of others theme. In addition to the fear of contracting the virus oneself, nurses also noted the stress caused by the restrictions imposed by the pandemic. After long, exhausting shifts working with critically ill and dying patients, nurses suffered additionally from having to isolate themselves from their loved ones. The restrictions sub-theme also included comments related to the closing of businesses, such as restaurants, theaters, and sporting events, which many felt could have helped them to alleviate some of their work-related stress. An Australian study identified social distancing due to lockdowns as an extra-added stressor to the work-related difficulties caused by the pandemic, depriving nurses of their much-needed sources of social support [36]. This sub-theme was largely reflective of the time point for the study, as Michigan’s governor instituted a stay-at-home order on 23 March 2020 that was not partially lifted until 24 April [37].

The theme related to the infection, illness, and death of others has been well-documented in media reports [5,6,8], editorials [23,24], and research [19,22]. Almost 40% of responses were coded to this theme. Nurses describe patients with panicked expressions, struggling to breathe and fearing death [5]. Observing patients’ struggle and experiencing an unprecedented level of patient deaths have been identified in media reports [6,8]. The related feelings of helplessness and inadequacy captured as a sub-theme are also echoed in several of these reports that describe nurses struggling to maintain their composure and just carry on, doing the best they can to help their patients [5,6]. An Italian study reported that the COVID-related illness or death of a colleague was associated with worse mental health in a sample of frontline healthcare workers [19]. A qualitative study from China found that witnessing patients’ experiences was difficult for both nurses and physicians [22].

The workplace theme accounted for more than half of all response codes. Many of these concerned critiques, noting the failure of the workplace to meet the nurses’ needs regarding safety, scheduling, and education regarding the care of COVID patients. The lack of adequate training was emphasized among nurses who deployed from other workplaces and/or specialties, a condition noted in several reports [7,23] and as a factor in Chinese nurses’ poor psychological health [38]. Workplace concerns have been documented during the COVID-19 pandemic, identifying issues with care planning and resources [3]. Several comments in this category concerned changes in nurses’ work hours/employment due to the pandemic, which is in line with a study of Australian nurses [36]. Inadequate support from the workplace has been previously identified in nurses’ experiences with the H1N1 influenza [27] and Middle East Respiratory Syndrome (MERS) pandemics [28]. A systematic review of acute care nurses’ experiences during a pandemic identified three main organizational issues: protection and safety; knowledge and communication; and adequate leadership, staffing and policies related to emergency preparedness [35]. Comments related to each of these three issues were encompassed by our workplace theme. Even in the midst of a terrifying global pandemic, core aspects of workplace functioning (trust, communication, collaboration) remain relevant. Organizations and teams that are able to maintain these domains may mitigate many of the other stressors reflected in our study’s other themes.

A related, yet separate theme, concerned stress related specifically to PPE—its provision, guidelines for its use, and the physical discomfort of wearing it for long periods of time. Lack of or inadequate PPE has been chronicled in media reports [20], editorials [2,24,39], as well as in a growing number of scientific studies [12,15,36,40]. Studies from the U.S. [13] and Portugal [15] reported significant associations between lack of PPE and increased prevalence of mental health disorders in nurses. Comments related to the need to reuse masks and other protective gear were in line with the findings of a cross-sectional study among Australian nurses [36]. A qualitative study from China found that the necessity of constantly using protective gear was considered a burden by nurses and physicians [22].

Nearly a quarter of the comments concerned dealing with the unknown. During the time of the study in May of 2020, and even now, several months later, much remains unknown as to the characteristics, treatment, and repercussions of this highly contagious virus. In a review of both articles and media reports, Neto et al. [2] found that the virus, itself, is a source of uncertainty. The newness of the situation for nurses, specifically, has been described as a source of fear and stress [23]. In the early days of the pandemic, this was described as a struggle to keep up with care protocols that were changing on a daily basis [5,22].

The final theme, opinions/politics, encompassed only about 10% of the comments but almost 100% agreement between coders. Media reports have documented nurses’ frustrations over the failure of healthcare executives and politicians to do everything possible to keep frontline nurses safe and protected [7]. Nurses have expressed both amazement and anger at the government’s lack of preparation for the pandemic, comparing the provision of PPE for healthcare workers in other countries with the scarcity and insufficiency of supplies in the U.S. [5]. The politicization of wearing masks [41] and inaccurate data reporting from the federal government [42] have also been recognized in media reports. A study from China also identified lack of support for healthcare workers in media reports as a source of stress for nurses [38].

Previous research on nurse burnout, both before [9] and during the COVID-19 pandemic [14] identified caring for large numbers of patients, lack of stress management [9], and continuous contact with pain and suffering [14] as risk factors. All of these factors were described in the themes identified in this qualitative analysis, and they are in many ways magnified by the tremendous stress that nurses are experiencing in the care of COVID patients. The in-depth view of work stress provided by the current study underscores findings of prior studies that predict greater emotional exhaustion [14] and an increased need for psychological help [14,18] among nurses in the absence of an effective response to COVID-19. In a long-term perspective, prolonged stress caused by the pandemic could potentially lead to an increase in nurse turnover and even thoughts of leaving the profession.

### Limitations

While this qualitative analysis sheds light on sources of nurses’ experiences of stress early in the pandemic, the study has some limitations. First, participants were limited to nurse members of three large nursing organizations in a single U.S. state (Michigan), and results may not be generalizable to nurses in other states or countries. Nevertheless, many of the themes identified in this analysis are mirrored in studies from China [38], Italy [19], Australia [36], and Portugal [15], suggesting a certain universality to the stress that nurses are experiencing. Second, significantly more nurses who did not respond to the open question (n = 240) worked in inpatient care, compared to nurses who did respond to the question (n = 455). It is possible that non-respondents may have experienced different work-related factors that may have influenced their perceptions of stress. Nevertheless, the majority of respondents (over 65%) did write in responses, which limits non-response bias. Our study sample was also substantially larger than those of two previous qualitative studies that explored the perceptions of 20 nurses [21] and 13 nurses and physicians [22] in China. As in all qualitative studies, researcher bias may influence findings. However, analyses in the current study were validated through a third researcher who was not involved in the original coding.

## 5. Conclusions

Exploration of nurses’ perceptions of stress during the pandemic’s early phase provides important insight into the nature of nurses’ experiences and potential measures that healthcare institutions can take to mitigate nurses’ stress. Providing nurses with adequate personal protective equipment is one concrete measure that can help to keep nurses safe and to alleviate their fear of becoming infected. Healthcare units should provide opportunities for nurses to discuss the stress they are experiencing, support one another, and make suggestions for workplace adaptations during this pandemic. Healthcare institutions and nurse managers need to recognize these sources of stress in order to identify potential organizational interventions to maintain nurses’ health, safety, and well-being.

## Figures and Tables

**Table 1 ijerph-17-08126-t001:** Overview of main themes and subthemes.

Main Themes	Sub-Themes	Definitions	% Endorsed ^†^ (Nurses)	% Agreement (Coders)
Exposure/Infection		Fear of the self being exposed to COVID-19 and becoming ill, actually being exposed and becoming ill, or fear of passing virus onto others	29.67%	94.30%
	Restrictions	Restrictions associated with the pandemic, such as social distancing and the closing of businesses	7.03%	
Illness/Death		Infection, illness, and death of others, usually patients, coworkers, or loved ones	38.90%	92.76%
	Inadequacies	Feelings of inadequacy and helplessness, especially in relation to their patients’ condition and treatment	6.15%	
Workplace		Work-related problems, including relationships with coworkers, perceived workplace administrative failings, and failure to provide supplies and training	51.21%	95.83%
PPE/Supplies ^‡^		Stressors related to PPE/Supplies, including dearth of PPE, unclear guidelines, and physical discomfort related to wearing PPE	21.98%	96.27%
Unknowns		Dealing with unknowns, including the changing understanding of viral symptoms and potential surges, job/financial security, etc.	22.64%	92.76%
Opinions/Politics		Family/community opinions related to COVID-19 and politicization of the pandemic. Includes dealing with perceived failings of state/federal administration, seeing false information spread, etc.	9.67%	99.34%

^†^ Sum of percentages is greater than 100%, as responses could be coded into more than one category. ^‡^ PPE: Personal protective equipment.

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
