# Peer review of "Nurse Reports of Stressful Situations during the COVID-19 Pandemic: Qualitative Analysis of Survey Responses"

_ijerph, 2020, doi:10.3390/ijerph17218126_

Round 1

Reviewer 1 Report

Nurse Reports of Stressful Situations During the COVID-19 Pandemic:

Qualitative Analysis of Survey Responses

This study addresses a topic of potential interest to readers: to assess: nurses’ experiences and well-being during the COVID pandemic crisis in order to identify risk groups for ill health and potential sources of organizational intervention.

The stated aims of the study were to explore perceptions of the most salient sources of stress in the early stages of the coronavirus pandemic in a sample of U.S. nurses.

The manuscript reports results of a cross-sectional online survey conducted among a sample of 695 U.S. nurses in May 2020. The report is well organized and clearly written. The study has moderate potential impact. It clearly identifies the gap in knowledge justifying the study, i.e., the importance of studying the experiences of frontline nurses to 91 inform effective workplace and national responses during future healthcare crises. Results of the study are not generalizable, although results are consistent with similar studies in other settings. The topic is appropriate to the Journal in that it focuses on an issue relevant to public and environmental health.

One minor comment is included for the authors’ consideration. On line 276, I suggest you replace one-third with two-thirds.

For the many reasons mentioned above, I recommend acceptance with minor revision.

Author Response

Our thanks to the three reviewers for their thoughtful and constructive comments. We have revised our paper based on their input and we believe it has helped to clarify and strengthen the paper’s message. Following is a point-by-point response to each comment in italics.

Reviewer #1

This study addresses a topic of potential interest to readers: to assess: nurses’ experiences and well-being during the COVID pandemic crisis in order to identify risk groups for ill health and potential sources of organizational intervention.

The stated aims of the study were to explore perceptions of the most salient sources of stress in the early stages of the coronavirus pandemic in a sample of U.S. nurses. 

The manuscript reports results of a cross-sectional online survey conducted among a sample of 695 U.S. nurses in May 2020. The report is well organized and clearly written. The study has moderate potential impact. It clearly identifies the gap in knowledge justifying the study, i.e., the importance of studying the experiences of frontline nurses to 91 inform effective workplace and national responses during future healthcare crises. Results of the study are not generalizable, although results are consistent with similar studies in other settings. The topic is appropriate to the Journal in that it focuses on an issue relevant to public and environmental health. 

One minor comment is included for the authors’ consideration. On line 276, I suggest you replace one-third with two-thirds. 

For the many reasons mentioned above, I recommend acceptance with minor revision.

Response: Our thanks to the reviewer for positive and supportive comments. We have now replaced one-third with two-thirds on line 276 in the original manuscript (line 290 in the revision).

Reviewer 2 Report

The abstract clearly and accurately describes the content of the manuscript. The methodology should be described in detail and more information regarding the research tool are needed (who developed the questionnaire and the item on which all the analysis was based). Some more attention should be paid in the way that citations are presented in the text (for example in lines 60,63) and minor changes regarding the references in the text would improve the manuscript. In discussion there is enough reference made to previous work and a comparison with results of other studies, but not in a clear way. The discussion part could be improved. In general, the present manuscript contains interesting results for the readers.

Author Response

Our thanks to the three reviewers for their thoughtful and constructive comments. We have revised our paper based on their input and we believe it has helped to clarify and strengthen the paper’s message. Following is a point-by-point response to each comment in italics.

Reviewer #2

The abstract clearly and accurately describes the content of the manuscript. The methodology should be described in detail and more information regarding the research tool are needed (who developed the questionnaire and the item on which all the analysis was based).

Response: The methodology used in the qualitative analysis is described in detail in sections 2.6-2.9, which explain the data analytical process, theme development, coding, and analytical rigor, respectively. In the revised manuscript, we have expanded our description of the research tool and its development in section 2.2, Research Design. We cite our previous publication (Arnetz et al., Journal of Occupational and Environmental Medicine 2020) that reported quantitative findings from the questionnaire.

Some more attention should be paid in the way that citations are presented in the text (for example in lines 60,63) and minor changes regarding the references in the text would improve the manuscript.

Response: To the best of our knowledge, we have followed the journal instructions for citation formatting in both the text and in the reference list. If the reviewer would please provide more specific suggestions as to how we can improve citation presentation in the text, we will be happy to make those changes.

Reviewer 3 Report

The work is interesting but you must make some changes before acceptance, I think important:
1. The theoretical foundation cannot be based on a description of the facts of COVID. There must be a foundation in the bases of the descriptors indicated and to which the work refers: occupational health; work stress. This is practically not analyzed, except in the context of COVID. I think the work is fine, I only indicate that an adequate review has not been carried out on the subject, which lays the foundations of the theory of variables involved in burnout and stress in health personnel. It is limited to analyzing the concept in the context of COVID, but the rationale must be based on the relationship of these concepts prior to the pandemic. It is important, because variables that have been previously studied and that become more evident during the covid are analyzed.

This section should be substantially improved, for example, by referring to mandatory works that help lay the foundations of the study:

https://journals.copmadrid.org/ejpalc/art/ejpalc2018a13

https://www.mdpi.com/2077-0383/9/9/3029 

https://www.mdpi.com/2077-0383/8/3/286 

https://www.mdpi.com/1660-4601/17/14/5041 

  1. The sampling procedure is not clear and how qualitative data are collected, this part should be improved.
  2. The practical aspects should be pointed out, what are the repercussions of the work in practice. They are little marked in the wording.

Author Response

Our thanks to the three reviewers for their thoughtful and constructive comments. We have revised our paper based on their input and we believe it has helped to clarify and strengthen the paper’s message. Following is a point-by-point response to each comment in italics.

Reviewer #3

The work is interesting but you must make some changes before acceptance, I think important:
1. The theoretical foundation cannot be based on a description of the facts of COVID. There must be a foundation in the bases of the descriptors indicated and to which the work refers: occupational health; work stress. This is practically not analyzed, except in the context of COVID. I think the work is fine, I only indicate that an adequate review has not been carried out on the subject, which lays the foundations of the theory of variables involved in burnout and stress in health personnel. It is limited to analyzing the concept in the context of COVID, but the rationale must be based on the relationship of these concepts prior to the pandemic. It is important, because variables that have been previously studied and that become more evident during the covid are analyzed.

This section should be substantially improved, for example, by referring to mandatory works that help lay the foundations of the study:

https://journals.copmadrid.org/ejpalc/art/ejpalc2018a13

https://www.mdpi.com/2077-0383/9/9/3029 

https://www.mdpi.com/2077-0383/8/3/286 

https://www.mdpi.com/1660-4601/17/14/5041

Response: Our thanks to the reviewer for pointing out this important oversight in our original manuscript. We have now incorporated a section on a theoretical foundation in an occupational health and safety perspective in the Introduction, second paragraph. We also return to this perspective in the final paragraph of the Discussion, before Limitations.

  1. The sampling procedure is not clear and how qualitative data are collected, this part should be improved.

Response: In section 2.3 that describes the study participants, we have added the following text, highlighted in red: “During the survey period, 695 responses were collected. Of these, 455 nurses also responded to the open-ended question (65.47%) and it is these nurses’ responses that are examined in the current study.

  1. The practical aspects should be pointed out, what are the repercussions of the work in practice. They are little marked in the wording.

Response: We have now added an additional paragraph at the end of the Discussion, right before the Limitations paragraph. We discuss our findings in the context of an occupational health and safety perspective, and we mention the ramifications of the enormous stress described by the nurses in our sample. The possible repercussions include increased emotional exhaustion, need for psychological help, and even thoughts of turnover and leaving the nursing profession.

Round 2

Reviewer 3 Report

The authors have made the indicated changes.